# The relevance of rock shape over mass—implications for rockfall hazard assessments

Andrin Caviezel [1✉], Adrian Ringenbach[1], Sophia E. Demmel[1], Claire E. Dinneen [1], Nora Krebs[1], Yves Bühler [1], Marc Christen[1], Guillaume Meyrat[1], Andreas Stoffel[1], Elisabeth Hafner [1], Lucie A. Eberhard[1], Daniel von Rickenbach[1], Kevin Simmler[1], Philipp Mayer [2], Pascal S. Niklaus [2], Thomas Birchler[2], Tim Aebi[2], Lukas Cavigelli [2], Michael Schaffner[2], Stefan Rickli [2], Christoph Schnetzler[2], Michele Magno[2], Luca Benini[2] & Perry Bartelt[1]

The mitigation of rapid mass movements involves a subtle interplay between field surveys, numerical modelling, and experience. Hazard engineers rely on a combination of best practices and, if available, historical facts as a vital prerequisite in establishing reproducible and accurate hazard zoning. Full-scale field tests have been performed to reinforce the physical understanding of debris flows and snow avalanches. Rockfall dynamics are - especially the quantification of energy dissipation during the complex rock-ground interaction - largely unknown. The awareness of rock shape dependence is growing, but presently, there exists little experimental basis on how rockfall hazard scales with rock mass, size, and shape. Here, we present a unique data set of induced single-block rockfall events comprising data from equant and wheel-shaped blocks with masses up to 2670 kg, quantifying the influence of rock shape and mass on lateral spreading and longitudinal runout and hence challenging common practices in rockfall hazard assessment.

[1] WSL Institute for Snow and Avalanche Research SLF, Davos, Switzerland. [2] ETH Zurich, Integrated Systems Lab IIS, Zurich, Switzerland. ✉email: caviezel@slf.ch

Two key factors in an accurate quantification of rockfall risk are realistic estimates of the release conditions and a sound evaluation of possible block propagation trajectories. Geologic, on-site, or remotely conducted studies define the location and spatial distribution of rock-face instabilities and, ideally, constrain the possible release scenarios in terms of volume, block mass, and shape as well as recurrence probability[1–6]. Three dimensional, numerical rockfall tools are then applied to determine propagation distances to assess rockfall hazard[7–13]. Once the constitutive parameters governing the rock-ground interaction are set for the respective model, the numerical simulations provide objective, spatially inclusive information on the relevant parameters of interest such as runout distances, jump heights, and kinetic energies as a function of the digitised terrain. While the awareness of rock shape dependence on rockfall trajectory behaviour is well established[14], only the advent of available computational means to incorporate complex shapes has triggered renewed interest in accurate size and shape treatment in rockfall hazard assessments[12,15–23] but equally its implications on numerical schemes[11–13,24].

Establishing an experimental foundation underpinning the non-smooth kinematics of rockfall motion presents the geohazard science and engineering community with many special challenges. The measurement techniques used to capture the essentially smooth propagation velocities of fluid-type natural hazards (see for example ref. [25] for debris flows or refs. [26,27] for avalanches) cannot be employed because they do not have the spatial

and temporal resolution to capture the sudden, short-duration impact phenomena governing rockfall motion. Several experimental studies of induced rockfall trajectories have been conducted[9,28–31,31–36]. Most of these studies focus on small numbers of rocks and/or rebounds, that is limited falling distances. Recent studies pave the way to a more complete and exhaustive experimental coverage of single-block experiments on a full slope scale[37,38] expanding experimental techniques for complete trajectory reconstructions similar to earlier pioneering studies[30,34]. These initial studies demonstrate the possibilities of new experimental techniques, they still do not have the statistical basis, i.e., number of experiments, range of rock types, etc., to quantify energy dissipation rates and therefore calibrate modelling tools.

While the larger part of a rockfall motion is represented as a series of oblique throws that follow ballistic trajectories with undisturbed rigid-body rotations, the complex rock-ground interaction leads to local and discrete energy transfer mechanisms. This makes the rockfall problem unique in natural hazards mitigation. Unlike other gravitationally driven hazards such as debris flows, or rock and snow avalanches, energy dissipation mechanisms in the rockfall problem cannot be smoothed by modelling statistical averages of thousands of particle interactions with the ground and/or other particles. Accurate modelling of the rock-ground interaction requires understanding the dynamic response of the ground loaded by a single, spinning, and complex-shaped rigid body. Motion patterns such as sliding, slip-free rolling (and combinations) increase the complexity and hence the selection of constitutive parameters governing rebound and energy dissipation.

This study presents an unprecedented dataset of induced single-block rockfall events. In terms of data volume, systematic sampling, and consistent boundary conditions, the measurements surpass existing rockfall datasets. Fusion of external and in situ measurement methods[39,40] enable exhaustive trajectory reconstruction yielding information not only over the complete flight path including parameters of interest such as translational velocity vectors, angular velocities, ballistic jump heights, and lengths but equally reveal insights in impact kinematics and energy dissipation mechanisms. The gathered data can thus serve as a unique calibration basis for numerical rockfall models.

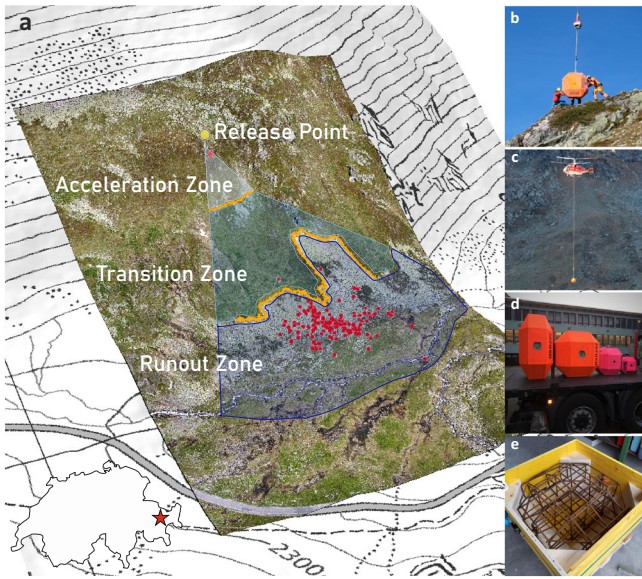

**Fig. 1 Aerial test site overview and experimental impressions of helicopter-aided single-block rockfall experiments. a** Perspective overview of the rockfall test site Chant Sura located on the Flüelapass, Switzerland, with its geographic location depicted in the lower-left corner. A UAS-derived orthophoto is draped over the corresponding digital elevation model on top of a regular swisstopo map (Source: Federal Office of Topography swisstopo). The release platform is marked with a yellow pin. The ensemble of deposited rocks is indicated with red markers. The acceleration zone above the cliff, transition zone between cliff and scree line, and runout zones/scree field are labelled. **b** Reception of an EOTA$_{221}^{2670kg}$ by the ground crew at the release platform. **c** Slinging of a EOTA$_{111}^{2670kg}$ rock by a Kamov KA32 A12 back to the release platform situated at 2380 m.a.s.l. **d** A subset of the used rocks ready for transportation: EOTA$_{221}^{2670kg}$ and EOTA$_{111}^{2670,800,200kg}$ from left to right. **e** Steel reinforcement cage of a wheel-shaped EOTA$_{221}^{200kg}$ block to ensure maximal ruggedness and lifetime.

## Results

**The Chant Sura experimental campaign.** The presented Chant Sura experimental campaign (CSEC) consists of data collected over 12 individual days spread over the snow-free seasons of 2017–2019 at the Chant Sura experimental site (46.74625°N, 9.96720°E) located on the Flüelapass, Switzerland (Fig. 1). Test samples are perfectly symmetric, man-made test EOTA blocks, the norm rock of the European Organization for Technical Assessment used in standardised rock fence testing procedures in official European Technical Approval Guidelines[41]. Heavily reinforced concrete blocks with weights of roughly 45, 200, 800, and 2670 kg—a subset of which are depicted in Fig. 1b—are cast and repeatedly released from a hydraulic platform located at 2380 meters a.s.l. (see insets of Fig. 1a). Initial conditions vary only slightly in rock orientation on the platform, positional deviations are negligible. The trajectory is initialised once the platform tilt allows the rock to slide off. Directional deviations of the individual trajectories are predominantly governed via geometrical configurations of the first impacts. A previous small rock experimental campaign[42] identified equant and compact platy rocks as the worst-case scenario drivers. Thus, these two form variants of the EOTA rock are used, the equant EOTA$_{111}$ with

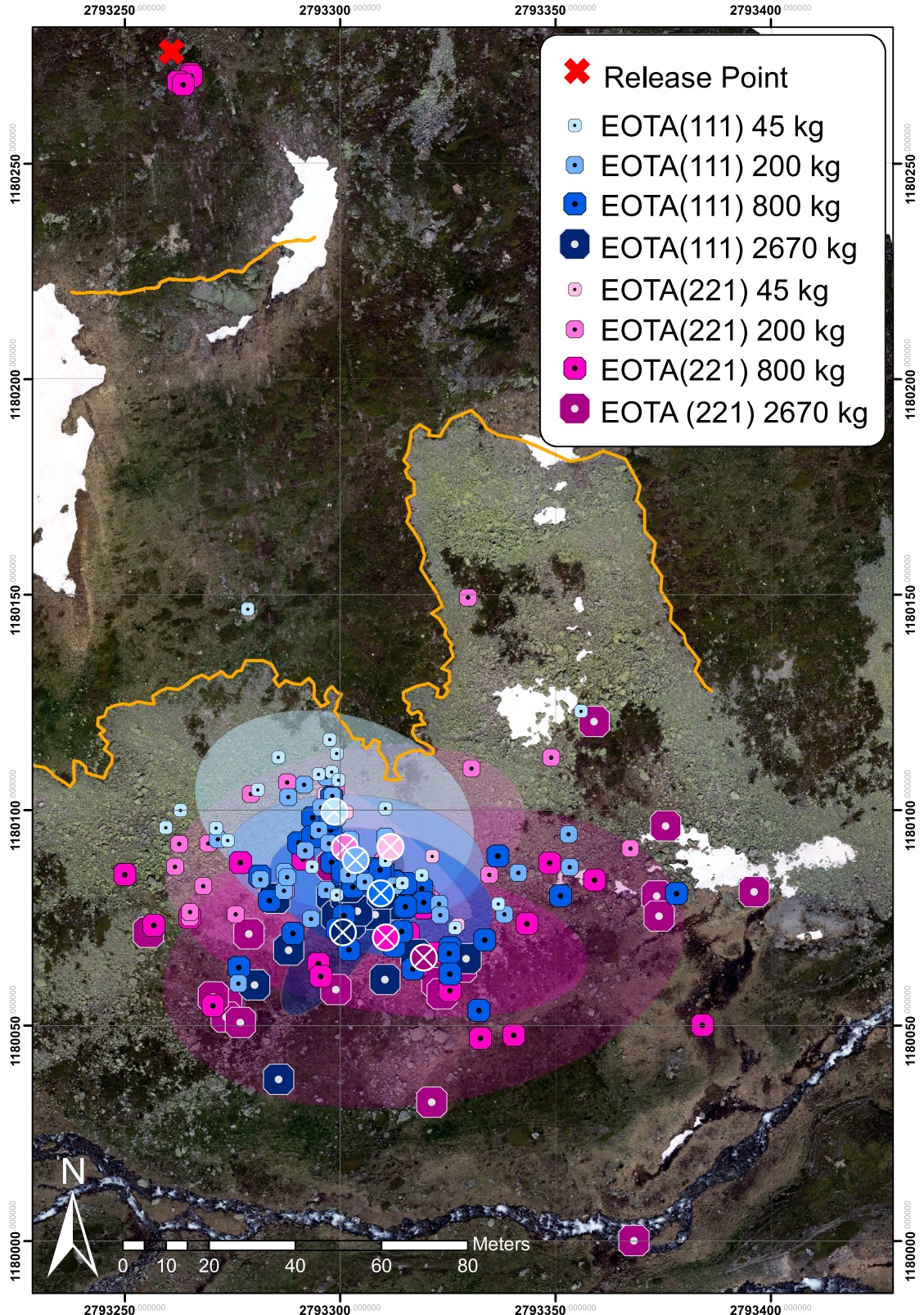

**Fig. 2 Complete set of 183 deposition points of the Chant Sura Experimental Campaign.** Blue markers represent equant EOTA$_{111}$ rocks varying from 45 kg (light blue) to 2670 kg (dark blue). Wheel-shaped EOTA$_{221}$ deposition points are shown in magenta with masses from 45 kg (light magenta) to 2670 kg (dark magenta). The respective centres of mass for each rock category is indicated with a crossed circle in the same colour together with its first standard deviation ellipse.

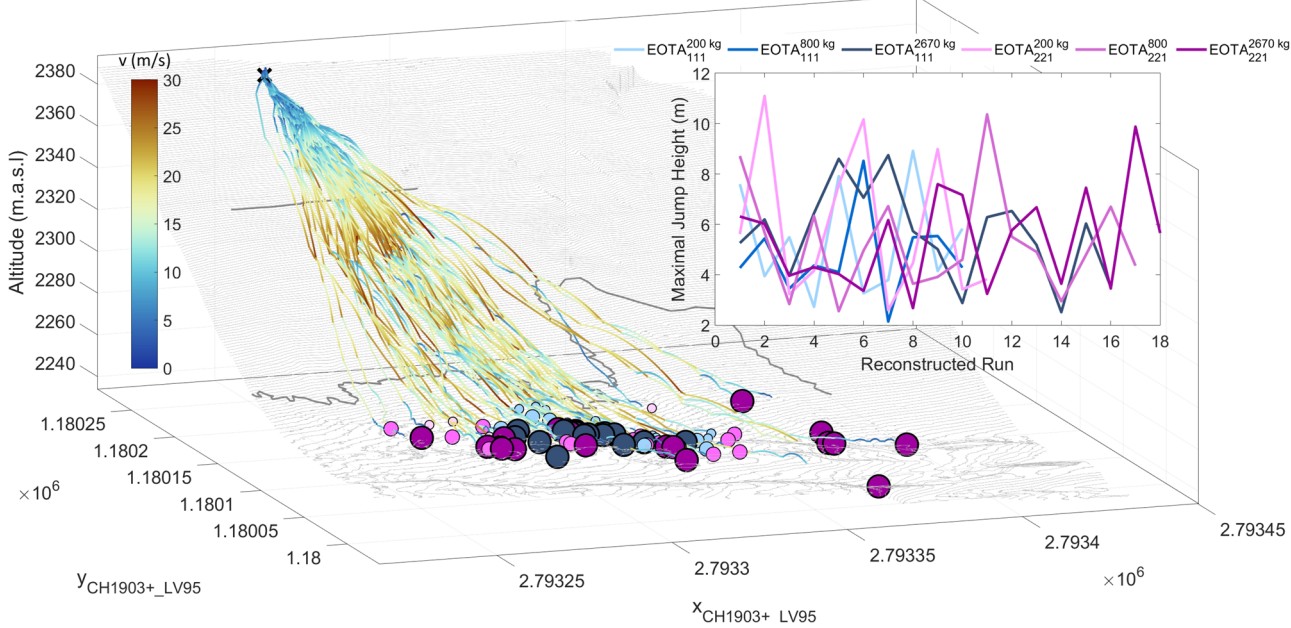

**Fig. 3 Complete set of 82 reconstructed rockfall trajectories of the CSEC and their maximal jump heights.** Complete set of 82 reconstructed trajectories of the CSEC. The colour and size code of the deposition points is identical to Fig. 2: Blue markers represent equant $EOTA_{111}$ rocks varying from 45 kg (light blue) to 2670 kg (dark blue). Equivalently, the wheel-shaped $EOTA_{221}$ deposition points are represented with masses from 45 kg (light magenta) to 2670 kg (dark magenta). The translational kinematics are visualised with a velocity scaled, perceptually uniform, colouring[48] of the trajectories. The inset corroborates the uniform kinematic behaviour across the shape and weight classes with similar overall maximal jump heights of each trajectory usually occurring at the cliff jump (upper thick grey line) as can also be derived from similar spatial velocity distribution of the trajectories.

three identical axis lengths and the wheel-shaped $EOTA_{221}$, where the wheel diameter is twice as long as the shortest axis.

The acceleration zone above the upper orange line in Figs. 2 and 3 has soil characteristics of dwarf shrubs while the transition zone shows typical characteristics of alpine meadow interspersed with rocks with a slope inclination between 20–60 degrees. The flat runout consists of a slightly dipping, rough scree field. Two prominent spatial hallmarks are the nearly vertical cliff located in the upper part of the slope and the scree line. Both are outlined with thicker lines in Figs. 1, 2, and 3. The test rocks are equipped with sensor nodes moving with the rock in what is mathematically termed the Lagrangian reference frame to track parameters of interest in situ. The deployed StoneNode v1.1-3[43], a dedicated inertial measurement unit (IMU), mounted in the rock's centre of mass records accelerations up to $400\,g$ and angular velocities up to $4000\,°/s$ at a data acquisition rate of $1\,kHz$. For a detailed presentation of the methodology consider[37] and the references therein. Rock transportation is ensured via Airbus H125, H225, or Kamov KA32 A12 helicopters depending on the rock masses (see Fig. 1c). The induced rockfall events are recorded via external static, so-called Eulerian, measurements such as high-resolution videogrammetry for optical projectile tracking. Camera constellations vary from single-camera setup up to static stereo-graphic videogrammetry via three spatially separated RED EPIC-W S35 Helium cameras recording 8K video footage consisting of a 25 frames per second image stream with an image resolution of $8192 \times 4320$ pixels. Most reliable high-speed synchronisation is achieved via a Tentacle Sync Lock-it set enabling jitter-free frame-wise temporal pairing of the image stream. High-accuracy differential GNSS handheld receivers such as the Trimble GeoXH and Stonex S800 are used to measure endpoint and scarring locations on the slope with an accuracy significantly below the extent of the deposited block.

For each experimental day, a high-resolution digital surface model (DSM) is generated pre- and post-experimentally via aerial

remote sensing using various unmanned aerial systems such as DJI Phantom 4 Pro, DJI Phantom 4 RTK, or the eBee+ equipped with 20–24 MP cameras. Flight planning is achieved with the respective software tools to ensure precise navigation on steep slopes and sufficient image overlap, set to 80% forward and 60% side overlap. Flight altitude above ground level averages around 75 m and a mapping area of $0.2\,km^2$ is covered. Roughly 500 images were taken during each mapping job, yielding a very high point density of ~600 points/$m^2$. To avoid systematic offsets between pre- and post-experiment flights, ground control points were distributed for the absolute reference orientation and mapped using a differential GNSS with tri-axial accuracies of 2–5 cm. The obtained UAS imagery is processed using the latest AgiSoft PhotoScan Pro v1.4.3-1.6.3, a commercial software extensively used in the UAS community. The photogrammetric workflow finally provides a DSM resolution of 5 cm and altitude uncertainties of ±3 cm, which allows the scaring mark detection in the corresponding differential DSM.

Figure 2 shows the complete set of 183 deposition points of the CSEC. Blue markers represent equant $EOTA_{111}$ rocks varying from 45 kg (light blue) to 2670 kg (dark blue). Wheel-shaped $EOTA_{221}$ deposition points are shown in magenta with masses from 45 kg (light magenta) to 2670 kg (dark magenta). The respective centres of mass for each rock category are indicated with a crossed circle in the same colour together with its first standard deviation ellipse for all the rocks reaching the runout zone.

**Four-dimensional trajectory reconstruction.** In total 82 rock trajectories were reconstructed in four-dimensional space— 3 spatial coordinates and time—visualised in Fig. 3. Reconstruction is performed in an ideal case via matching temporal information about the impact and lift-off taken from the sensor streams combined with scarring locations taken from differential

UAS surface models. The colour and size code of the deposition points are the same as in Fig. 2, with blue markers representing the deposition points of equant EOTA$_{111}$ rocks varying with darker shades for heavier rocks. Equivalently, the deposition points of wheel-shaped EOTA$_{221}$ vary from light to dark magenta according to their weight class. The flight paths in Fig. 3 are colour-coded by translational velocity, showcasing maximal velocities of roughly 30 m/s normally reached after the longest airborne free-fall phase after the cliff. All subsets consistently reach maximal velocities of 30 m/s in the cliff region. The translational kinematic behaviour is rather uniform across the shape and weight classes. Maximal jump heights are 8.5 m for all EOTA$_{111}$ blocks, while the EOTA$_{221}$ samples feature maximal jump heights of 11.1/10.4/9.8 m for the 200/800/2670 kg classes plotted as an inset of Fig. 3. Often, the rocks descend close to the ground, skimming the surface and spinning rapidly, covering the 250 m between release and deposition in less than 25 s. Maximal resultant angular velocities of 1000–5000 °/s (5.56$\pi$–27.78$\pi$ rad/s) are measured. The insets highlight the comparison of angular velocities for both shape classes. During a typical descent, a rock increases its angular and translational velocity during the acceleration/stabilisation phase. On this slope, this happens in the pre-cliff zone above the upper thick grey line in Fig. 3. Here, wheel-shaped rocks tend to stabilise around their largest moment of inertia—if not stopped immediately due to a landing on their flat side at low speed, see uppermost deposition points in Fig. 2 and hence have been omitted in the calculation of the deviational ellipses. Decreasing maximal rotational speeds with an increased moment of inertia along with the almost exclusive uni-axial rotations around the principal for wheel-shaped rocks are discernible both in the insets of Fig. 3 as well as in Fig. 4.

Of interest for rockfall hazard assessment is that the total spreading angle of all trajectories is 38 degrees. The spreading angle of the equant rocks reduces from $\phi_{111}^{45kg} = 32°$ to $\phi_{111}^{200kg} = 23°$, $\phi_{111}^{800kg} = 27°$ as low as $\phi_{111}^{2670kg} = 13°$. wheel-shaped rocks display inverse spreading behaviour with increasing spreading angles from $\phi_{221}^{200kg} = 30°$, $\phi_{221}^{800kg} = 32°$ to $\phi_{221}^{2670kg} = 36°$. Moreover, the wheel-shaped rocks determined the maximum spread of the trajectories, while the equant shaped rocks show only marginal spreading. The spreading angle appears to be defined immediately after release; that is, many of the wheel-shaped rocks followed straight trajectories once uprighted and stable. This is a strong indication that the initial wobble phase immediately after release determines rock spreading. This fact could be exploited in rockfall hazard assessment.

**StoneNode data—Lagrangian methods gather inside information**. The occasional sensor failure in such a highly dynamic environment is inevitable. The StoneNode sensor proved itself as highly rugged equipment. Mechanical failure of the internal accelerometer lead to data loss in five runs, a wrong compiler setting on the gyroscopic sensitivity programming lead to 29 recorded runs with compromised data quality. The main limitation for sensor data availability in the individual runs, was the number of available sensors to equip all rocks, as they were prototypically developed during the first part of the CSEC. The overall sensor stream availability amounts to 66% for accelerometer data and 63% for gyroscope data. Figure 4a compares the gyroscope sensor output from both shape classes across the investigated weight categories. Visible is the slope-specific bell curve evolution of angular velocity. The decreasing absolute resulting rotational speed with an increased moment of inertia as well as the almost exclusive uni-axial rotations around the principal axis for wheel-shaped rocks are predominant features. To

scale the slope specificity for a given weight or shape class to a single measure the extrapolated estimated mean rotational speeds for different mass classes are shown in Fig. 4c. While 10 kg rocks are expected to rotate up to 4000 °/s (11 Hz) it decreases to 130 °/s for a 100 t block (0.36 Hz). The empirical power-law

$$\bar{\omega}_{111} = a_{111} \cdot m^{-0.37 \pm 0.06} \quad \text{and} \quad \bar{\omega}_{211} = a_{221} \cdot m^{-0.44 \pm 0.12} \quad (1)$$

with the scaling parameters $a_{111} = 9278$ [° s$^{-1}$kg$^{0.37}$] $a_{221} = 12480$ [° s$^{-1}$kg$^{0.44}$] and mass $m$ can provide a calibration basis across the entire mass spectrum for both shape classes for this slope. Here, the shape specificity vanishes within the uncertainty, which corresponds to the uniform kinematic behaviour across shape classes. Equant rocks show tumbling behaviour, meaning low frequency, large amplitude oscillation in the minor axis extracted as its Fourier transform spectra in Fig. 4d. wheel-shaped rocks on the other hand feature a higher frequency, faster wobbling around their minor axis, rather insignificant with respect to the overall rotational behaviour. Figure 4e visualises the energy ratio development upon descent for the chosen runs of Fig. 4a. The maximal rotational energy ratio of 40% can serve as the upper limit estimation for energy threshold estimation for rockfall barriers.

The use of Lagrangian methods, i.e., in situ instrumentation, allows for comparably effortless extraction of key parameters once the availability of rugged sensor nodes covering the needed parameter ranges is given. Above all, matching angular velocities between simulation results and experiments is a seal of quality for simulation code that relates to rigid-body approaches and needs to calculate rock-ground interactions with its full geometrical and kinematical complexity. The presented rotational magnitudes and their empirical law may serve as a benchmark for calibration routines.

**The stiletto effect: area-dependent energy dissipation**. The 82 reconstructed trajectories consist on average of eighteen parabolic sections, and a total of 1394 impacts can be analysed across all the weight and shape classes for translational degrees of freedom, 656 impacts additionally feature valid gyroscope information. The parabolic sections resulting from impact reconstruction always range from specified lift-off to subsequent impact location at the differential DSM. All parameters of interest (POI) such as position, kinematic variables, StoneNode readings such as angular velocities and acting accelerations are extracted at any given reconstruction point and allow to derive secondary variables such as translational and rotational energy. Impact analysis consists of extracting POI at the endpoint of one parabolic section and the lift-off POI of the subsequent parabolic section. Each impact corresponds thus to a given scar length from impact point to lift-off point. The total energy is specified as $E_{kin} = E_{trans} + E_{rot}$ and reduced to $E_{trans}$ in case of gyroscope sensor failure. The average soil-impact energy dissipation normalised per rock mass amounts to $\Delta E_{kin}^{kg} = -0.042 \pm 0.005$ kJ/kg independent of rock shape and form. When additionally normalising to the penetrating cross-section area as shown in Fig. 5a, an empirical mass and area normalised energy dissipation power-law emerges

$$\Delta E_{m\&a} = b \cdot m^{-0.7 \pm 0.1} \quad (2)$$

with $b = -9$ [kJ kg$^{-0.3}$ m$^{-2}$] as depicted in the insets of Fig. 5a. Figure 5b depicts the absolute mass and cross-section normalised angular velocity change per impact. The deducted empirical power-law follows a $m^{-1.49 \pm 0.02}$ dependency for cubic EOTA$_{111}$ rocks, and $m^{-1.2 \pm 0.1}$ for platy EOTA$_{221}$ rocks, respectively, as depicted in the insets of Fig. 5b. The extrapolation and validity of those empirical fitting laws to smaller and larger masses as well as their site-specificity must be verified in further studies. Future data sources can be further experimental campaigns conducted under similar, controlled conditions at different locations or the

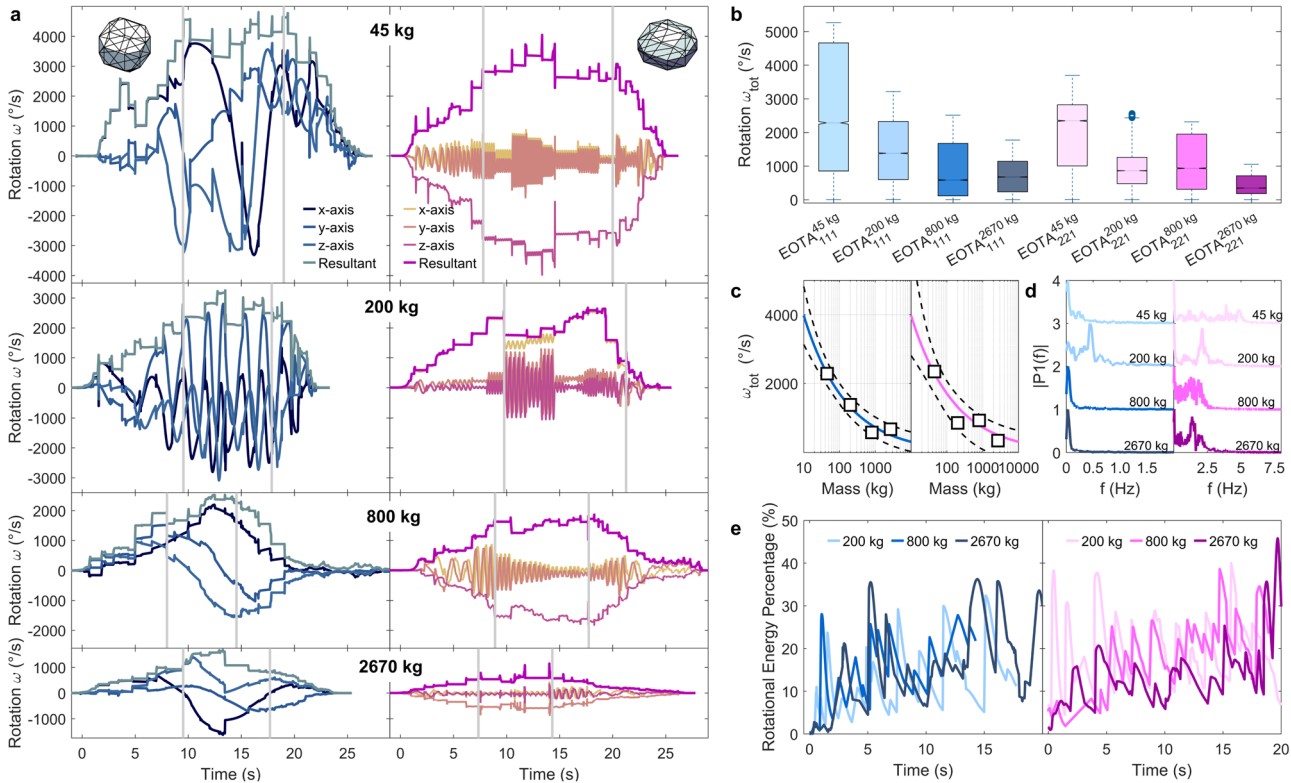

**Fig. 4 Gyroscope data streams across all shape and weight classes and their derived benchmark measures, such as median rotational relationship, wobbling frequencies, and energy ratios. a** Gyroscope data streams for all measured rock shapes and masses. Grey lines separate the acceleration zone above the cliff, transition zone between cliff and scree, and scree runout zones. Individual lines represent x-, y-, z-axis as well as its resultant angular velocities (thicker line). Decreasing maximal rotational speeds with an increased moment of inertia along with the almost exclusive uni-axial rotations around the principal axis for wheel-shaped rocks is visible, i.e. almost perfect overlap between the leading axis and resulting axis apart from the acceleration phase. Here, different colour shades represent different rock axis. **b** Summary boxplot—median values with standard whisker coverage of 99.3% ($2.7\sigma$)—of the StoneNode gyroscope output from the entire Chant Sura experimental campaign with its median rotational relationship as a possible benchmark indicator for unobstructed slopes plotted in (**c**). **d** Wobbling frequency components of descending rocks. **e** Rotational energy percentage of the presented runs in (**a**). Colour code for **b–c** is equivalent to Fig. 2: Blue markers represent equant rocks varying from 45 kg (light blue) to 2670 kg (dark blue), Wheel-shaped rocks are shown in magenta with masses from 45 kg (light magenta) to 2670 kg (dark magenta).

use of publicly available results of scientific literature that follow a similar data collection approach given the temporal information in the trajectory data is available. Figure 5d shows the shape-class-dependent angular velocity changes upon 1361 analysed impacts across the different weight and shape classes. Horizontal lines mark the maximal increase or decrease in angular velocity featuring the expected inertia dependency.

Figure 5c depicts the mass-normalised energy dissipation per impact with increasing impact velocities. The added lines represent exponential fits to the different weight and shape classes in the established colour code and serve as a guide to the eye. It confirms the intuition, that soil impacts of faster projectiles dissipate more energy as they generally penetrate further into the soil for equivalent soil conditions than smaller rocks. This is owing to the fact, that the compressibility of the impacted soil is limited to a velocity-dependent maximal penetration depth[44]. This maximal penetration depth is dependent on soil composition and impact configuration, meaning larger and faster rocks produce larger scars upon impact. The available travelling distance for momentum reversal therefore can reach an upper limit and scarring distance remain lower for slower projectiles. Analogously to the high-pressure loads of a person walking in stiletto heels, smaller rocks dissipate significantly more energy compared to larger ones owing to their smaller cross-section. This stiletto effect for rock-ground interactions scales with the travelling velocities as is depicted in Fig. 5c, similar to a running

person. It also remains shape-independent as the empirical fit lines group according to their weight classes. The presented data indicates that for specific load cases with small rocks, local peak energies may play a role leading to the so-called "bullet effect", the perforation of a rockfall protection mesh by an impact of a small block[45], and with rocks larger than 1 m³, the velocity dependence in energy dissipation needs only consideration starting from roughly 25–30 m/s. The occasional impact with positive $\Delta E_{m\&a}$ can be attributed to special rebound conditions—in terms of impact geometry and local impact conditions as for example on a downhill inclined smooth rock face in addition with a considerable potential energy intake when the impact happens on steep terrain. A major uncertainty in the reconstruction arises from impact/lift-off position placement and result in reconstructed velocity uncertainties of ±0.5 ms[−1][37]. Figure 5d plots the mass and area normalised absolute rotation change of every exploitable and its order of magnitude shifts of rotation changes for the different mass classes.

## Discussion

The two key physical problems that need to be solved to improve rockfall hazard assessment are: (1) determining how rocks laterally disperse from a given release source in natural mountain terrain, and (2) understanding the loss of translational and rotational kinetic energy during short-duration impacts with the ground. Energy dissipation during the ground interaction defines

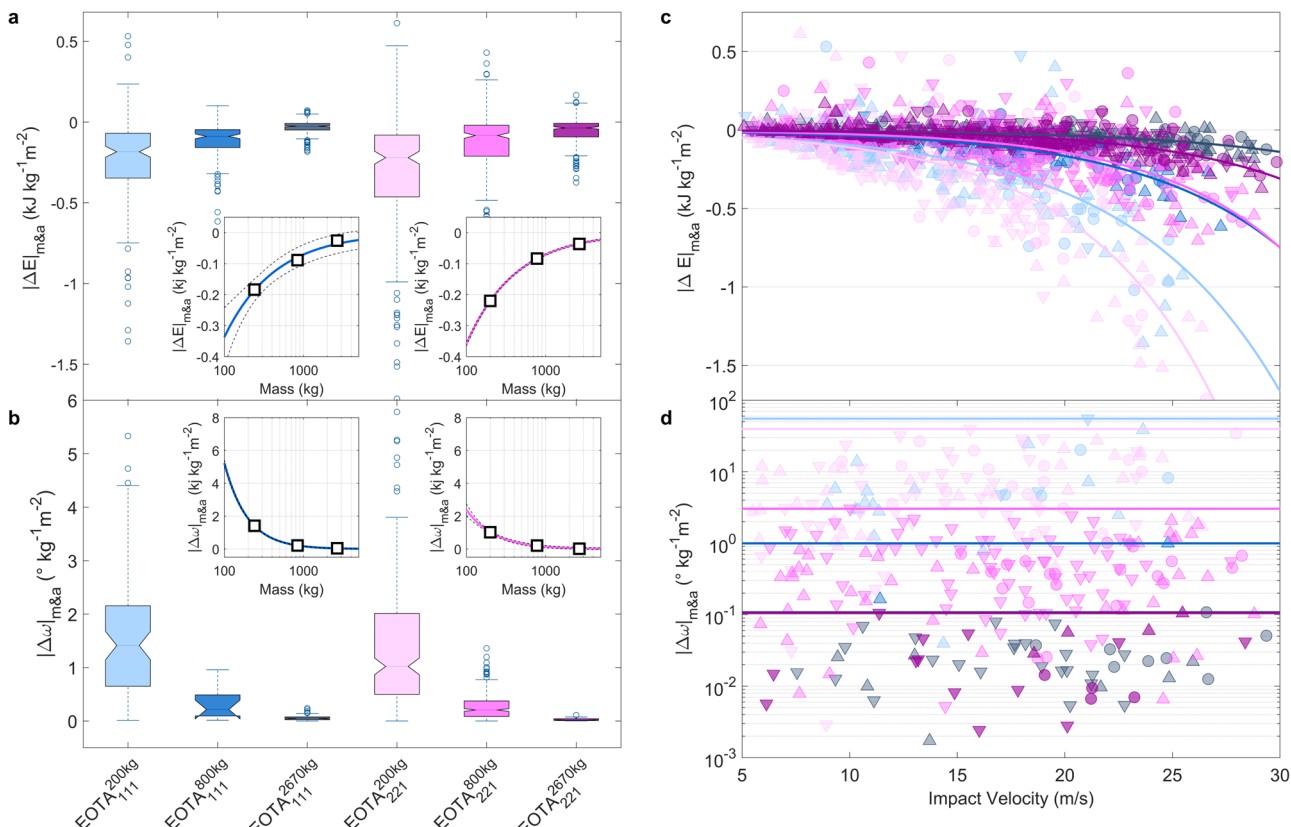

**Fig. 5 Mass and impact cross-section normalised energy dissipation and angular velocity changes for all analysed impacts. a** The stiletto effect in rockfall—mass and impact cross-section normalised energy dissipation boxplot for the three larger weight classes. Insets present empirical power laws for both shape categories with its $1\sigma$-prediction bounds. **b** Impact-induced changes in absolute angular velocity scaling with mass and surface area. Insets present empirical power laws for both shape categories with its $1\sigma$-prediction bounds. Both boxplots feature median values with standard whisker coverage of 99.3 % ($2.7\sigma$). **c** Velocity dependence of weight and impact cross-section normalised energy dissipation of 1394 analysed impacts for the three larger weight classes. The smaller the rock cross-section the more pronounced the effect—equivalently to a person running in stiletto high heels as opposed to sneakers. The lines represent exponential fits to the different weight and shape classes in the general colour code and serve as a guide to the eye. Upward facing triangles mark impacts in the acceleration zone, circles in the transition zones, and downward facing triangles in the runout zones. **d** Angular velocity changes upon 656 analysed impacts across the different weight and shape classes impact areas. Horizontal lines mark the maximal change in angular velocity per weight class. Equivalent color code as in Fig. 2 used.

rock speed, jump heights, and finally runout distance. Rockfall models must be able to correctly predict these values for accurate hazard assessments.

Here, we present the results of a multi-year experimental campaign resulting in an unprecedented, comprehensive data inventory of single-block-induced rockfall experiments. The significantly higher lateral spread of wheel-shaped rocks is striking and highly relevant for hazard assessments. Once upright, the wheel-shaped rocks appear to follow straight trajectories on wide, open slopes, unencumbered by minor slope changes perpendicular to their direction of motion. Fluid-type gravitational movements, like debris flows and snow avalanches, would react to these slopes, quickly following the line of steepest descent. The trajectories of wheel-shaped rocks resist this tendency, with the end effect that rocks can escape channels and terrain undulations, thereby increasing the width of the hazard zone.

We used the trajectories to quantify the change in rotational speed $\Delta\omega$ and change in translational kinetic energy $\Delta E$ during each ground interaction. We found that $\Delta\omega_{111} \propto 1/m^{0.83}$, $\Delta\omega_{221} \propto 1/m^{1.1}$, and $\Delta E \propto 1/m^{0.70}$. If we let the rock mass $m$ be approximated by a perfect sphere of radius $r$, then $m \propto r^3$. We find approximately,

$$\Delta\omega_{111} \propto \frac{1}{r^{2.4}} \ , \ \ \Delta\omega_{221} \propto \frac{1}{r^{3.3}} \ \ \text{and} \ \ \Delta E \propto \frac{1}{r^2} \quad (3)$$

This is a much stronger radius dependence as the $\Delta\omega \propto 1/r$ behaviour of a stick-slip-rolling transformation of an ideal sphere projected along a rough horizontal surface[46]. Frictional forces at the rolling interface apply a force and equivalent torque at the rock's centroid. Thus, the compilation of all the Chant Sura measurements indicates, that force tractions on the rock's surface control both the rotational and translational changes in kinetic energy and rock kinematics. Further analysis might focus on disentangling accelerating and decelerating behaviour and potentially identifying an optimal rock radius concerning energy accumulation and dissipation during impacts.

This result underscores once again the importance of rock shape. In real applications, for irregularly shaped rocks, the radius $r$ changes from impact to impact depending on the terrain slope and rock orientation; the traction force changes with the properties of the ground material. The difference between the measurements and this ideal result should now be exploited to quantify traction forces for irregularly shaped rocks impacting different ground types. The results presented here provide a detailed overview of the kinematic behaviour of two distinct rock shape classes up to $1\,m^3$ volume on the test site Chant Sura. While translational kinematics present themselves uniform across all shape and weight classes, strong shape dependence is found on the lateral spread and rotational axis stabilisation. Generalised,

site-specific relationships for rotational and energy dissipation behaviour are postulated and may serve as calibration measures. Despite very high level of detail in the data, single impact configurations are not resolvable down to their exact impact configuration and location. Future sensor node development coupling several sensor nodes to a fixed configuration might overcome the reconstruction limit given by error accumulation through integration of single IMU nodes and provide the missing information. The confirmation or adaptation of the proposed relationship for different site configurations will be of major interest for future work. Possible necessary modifications through topographic peculiarities such as couloirs or cliffs, the effect of forests, and different lithology remain of importance. Future studies aim towards rotational energy dissipation mechanisms upon impacts in general and during rock-barrier interaction in flexible rockfall barriers in particular.

The primary message of this work is therefore to incorporate shape effects in state-of-the-art models used for hazard zoning[13], defining hazard scenarios not only merely via block sizes but equally—if applicable—block shapes, that is incorporating a set of site-specific, realistic rock shapes in hazard assessments. As comparable datasets are scarce to non-existent, the now publicly available CSEC dataset[47] allows the geohazard community to re-evaluate many of the existing models, which have been calibrated based on oversimplified assumptions and data sources. It serves as an additional contribution to experimentally dissect rockfall propagation processes and could serve as a building ground for more complex impact models due to its high number of recorded individual impacts. It enables detailed future sub-studies of different contact models and energy dissipation mechanisms and the application of advanced machine learning algorithms as test ground for suitably tuned neuronal networks for the investigation of inertial measurement unit sensor data. At last, the dataset can serve as a calibration landmark for any numerical model and hence leads to more accurate hazard assessment and ultimately higher safety margins for societies in rockfall prone environments.

## Data availability

The Chant Sura Experimental Campaign dataset generated in this study has been deposited in the EnviDat repository under Caviezel, Andrin et al. (2020). Induced Rockfall Dataset #2 (Chant Sura Experimental Campaign), Flüelapass, Grisons, Switzerland. EnviDat. https://doi.org/10.16904/envidat.174.

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

# ARTICLE

37. Caviezel, A. et al. Reconstruction of four-dimensional rockfall trajectories using remote sensing and rock-based accelerometers and gyroscopes. *Earth Surf. Dyn.* **7**, 199–210 (2019).

38. Bourrier, F., Toe, D., Garcia, B. et al. Experimental investigations on complex block propagation for the assessment of propagation models quality. *Landslides* **18**, 639–654 https://doi.org/10.1007/s10346-020-01469-5 (2021).

39. Niklaus, P. et al. Stonenode: A low-power sensor device for induced rockfall experiments. In *IEEE Sensors Applications Symposium (SAS)*, 1–6 (2017).

40. Gronz, O. et al. Smartstones: A small 9-axis sensor implanted in stones to track their movements. *CATENA* **142**, 245–251 (2016).

41. ETAG 027. *Guidline for european technical approval of falling rock protection kits.* https://www.eota.eu/etags-archive (2013).

42. Caviezel, A., Bühler, Y., Christen, M. & Bartelt, P. *Induced rockfall dataset (small rock experimental campaign), tschamut, grisons, switzerland.* https://www.envidat.ch/dataset/experimental-rockfall-dataset-tschamut-grisons-switzerland (2018).

43. Caviezel, A. et al. Design and evaluation of a low-power sensor device for induced rockfall experiments. *IEEE Trans. Instrum. Measure.* **67**, 767–779 (2018).

44. Lu, G. et al. Modelling rockfall impact with scarring in compactable soils. *Landslides* **64**, 41 (2019).

45. Spadari, M., Giacomini, A., Buzzi, O. & Hambleton, J. P. Prediction of the bullet effect for rockfall barriers: a scaling approach. *Rock Mech. Rock Eng.* **45**, 131–144 (2012).

46. Cornwell, P. J., Pierre Beer, F. & Russell Johnston, E. In *Vector Mechanics for Engineers: Dynamics,* vol. 2. (McGraw-Hill, 2010).

47. Caviezel, A. et al. Induced Rockfall Dataset #2 (Chant Sura Experimental Campaign), Flüelapass, Grisons, Switzerland. *EnviDat* https://doi.org/10.16904/envidat.174 2020.

48. Crameri, F., Shephard, G. E. & Heron, P. J. The misuse of colour in science communication. *Nat. Commun.* **11**, 5444 (2020).

## Acknowledgements

We thank the municipality of Zernez, Switzerland, for their continuous support and permission to conduct experiments on the Chant Sura test site. Special thanks go to HeliBernina, Heliswiss International AG, and Heli Air AG for their repeated precision slinging work. We thank Matthias Paintner and Kevin Fierz for their support and guidance with the 8K videogrammetry.

## Author contributions

A.C. conceived the experiment. A.C., A.R., S.E.D., Y.B., M.C., G.M., A.S., E.H., L.A.E., D.v.R., K.S. and P.B. carried out the experiments. P.M., P.S.N., T.B., T.A., L.C., M.S., S.R., C.S. and M.M. developed StoneNode Sensors and analysed sensor data. A.C., A.R., S.E.D., C.E.D. and N.K. analysed the data. A.C and P.B. wrote the manuscript with discussions and improvements from all authors. L.B. and P.B. supervised the work.

## Competing interests
The authors declare no competing interests.

## Additional information

 9