## [Peer Review File · Nature Communications]

REVIEWER COMMENTS

Reviewer #1 (Remarks to the Author):

This paper presents results from an extensive campaign of block propagation experiments. The experimental procedure and, in particular, the measurement of block trajectories and kinematics changes during interaction with the ground were done using advanced techniques. This paper is of interest for the scientific community in the field of rockfall hazard given the amount of data acquired and the quality of the measurements done. In addition, this data is freely available in an Envidat repository. For these reasons, I recommend publication of the paper.

I don't have specific comments as regards to the introduction and study site description since both sections are very well written and show the expertise of the authors as regards to this topic. However, I also think that several points have to be improved in particular concerning the description of the details of the measurements, results analyses and the discussion. For that reason, I recommend major revision of the paper.

The following major points have to be clarified / improved :

- 1) Experimental procedure : the initial conditions are not detailed although they are major causes of the propagation variability. These conditions should be detailed in terms of initial release height, block initial orientation, procedure used to initiate propagation. The variability of these conditions could also be quantified.
- 2) Measurements : end point and scarring point locations. The authors present a 5 cm resolution that is very low according to me given that these points have to be estimated by the operator. Could the authors provide information on this point ?
- 3) Measurements: the authors used topography, cameras, accelerometers and gyroscopic sensors. However, the role of the different measures in the analysis is not clearly described (for example, were the trajectories reconstructed in 3d using only the cameras or also using the locations of scarring points ?). Could this point be clarified ?
- 4) Figure 3 : I think that the time evolutions of angular velocities (upper rights) are examples for one block of each size. It may be specified. How variable are these time evolutions from one block to another ?
- 5) Equation 1 : it seems to me that this relationship was established using all data per rock type and rock mass. So, it is a « mean relationship ». Could the authors clarify the meaning of this « mean relationship » given that, in Fig. 3, it is shown that the angular velocity evolves along the trajectory of each block ?
- 6) Equation 2 : Could the authors specify how the kinetic energy changes were calculated. Only the translational velocities were used ? How did the authors determine the velocity « before interaction » and the velocity « after interaction » ? What is the definition of an « interaction » ?
- 7) Figure 4 and Figure 5 : all the plots of the statistical relationships (small graphs) are plotted for mass values ranging from almost 0 to more than 50000 kg although the larger mass of the rocks used is 2650 kg. I think that the range of the plot should be reduced to the range of the blocks released.
- 8) Figure 5 c : the positive values of the kinetic energy changes during impact suggest increases in kinetic energy. It can be explained if only translational velocities are used. This interesting point has to be discussed.
- 9) Figure 5 c : the changes in rotational velocities during interaction are almost not described. It could be interesting to try to interpret these changes, especially in relation to changes in kinetic energy during impact.

9) p.6 l 194-200 : I don't understand.

10) Conclusion : it must be specified that the relationships obtained are site-specific and should be confirmed on other sites ?

11) Conclusion : I agree that the paper emphasizes the « effect of block shape on propagation » but the effects shown in this paper could be recalled and summarized here ?

Minor points :

1) p.6 l.196 : "In an analogous"

2) Figure 4 b : typo

Reviewer #2 (Remarks to the Author):

The paper presents the results of an extensive rockfall testing campaign conducted at the Chant Sura site in Switzerland. I found the paper well written and containing a unique large data set of full scale rockfall testing results.

However, a more comprehensive analysis of the results and some further discussion about their application for hazard assessment purposes could have been provided.

The introduction summarises some of the numerous studies conducted over the last four decades in the scientific literature. However, some pioneering research on rockfall impact/rebound and energy loss at impact should also be considered.

The experimental set up and methodology for the reconstruction of trajectories have already been extensively presented in [37]. While this is one of the most interesting aspects of the proposed work, the full trajectory reconstruction of falling blocks by video photogrammetry (including the full set of dynamic parameters at impact and rebound) is not new, and it has already been achieved by several researchers in recent years (see Dewez et al., 2010; Mathon et al. 2010; Giacomini et al. 2012; Volkwein et al. 2018 to name a few).

Similarly, the generation of high-resolution DSM by aerial photogrammetry, and the post processing of data using a commercial software such as PhotoScan are becoming very common practice in remote sensing applications for rockfall analyses (see recent works published on the topic in various international journals in the field of remote sensing).

Recent work on the consideration of block shape has also been conducted on the numerical investigation of the shape effect on rockfall motion (see Ji et al. 2019, Yan et al. 2019). The tendency of wheel shaped rocks to stabilise around their largest moment of inertia has already been highlighted by previous researchers and observations about "stiletto effect", or simply stress concentration, related to the "bullet effect" is also not new (see [38] and related works).

I appreciate the significant number of tests conducted in the research and I acknowledge that it is, indeed, a very valuable and unique data set. However, the observations collected in terms of flight paths, associated velocities, and measured spreading angle are site specific. The empirical power law proposed for the angular velocity is, therefore, site dependent only for future numerical calibrations.

The authors conclude that the spreading angle definition after release could be exploited in rockfall hazard assessment, however, no methodology is proposed for this purpose.

Graphs included in Figure 3 are hardly readable. Similarly, Figure 4, results very difficult to read because of the small fonts used in the graphs. Additionally, its significance (StoneNode outputs are reported for the entire experimental campaign) is also quite unclear.

The presentation of the results is quite narrow and, given the extensive data set, a more thorough analysis could have been conducted.

Reviewer #3 (Remarks to the Author):

The work of Caviezel et al. studies the results of a block release experiment that was documented in detail (sensors inside the blocks, high-resolution video records, etc.). Although it is not the first publication on this subject by this scientific team, their experimental set up is rather original and allow gathering new information on the mechanisms at stakes during rockfalls. I find this work relevant for publication.

I welcome the effort of transparency and contribution to the scientific community by making available the data collected during this experiment.

-Introduction:

-Stat of the art: Some references are missing (see minor comments)

-The authors should be more specific about their contribution in rockfall hazard assessment (see minor comments)

-Experimental set up: OK

-Methodology used to analyze the results of the experiment: on the overall, it is OK. Some explanations are missing (see minor comments).

-Discussion: on the overall, it is OK (see minor comments).

Minor comments:

1) overall remark:

I appreciate the enthusiasm of the authors who want to improve practices in rockfall hazard assessment. However, I would like them to express more specifically their contribution to this scientific field, especially in the abstract and the introduction. Their work provides concrete elements to improve the calibration of propagation models (as they state in the discussion), which represents only one step among many in rockfall hazard assessment. Even in this specific field of assessing rockfall propagation, research is still very active on documenting other physical phenomena (fragmentation of blocks, interactions of blocks during propagation, etc.) which are not mentioned here.

2) Line 20: "Presently, there exists no experimental basis on how rockfall hazard scales with rock mass, size and shape": There is very scarce work on the subject. However, some works were recently published on the subject:

•Wegner, K., Haas, F., Heckmann, T., Mangeney, A., Durand, V., Villeneuve, N., & Becht, M. (2020). Assessing the effect of lithological setting, block characteristic and slope topography on the runout length of rockfalls in the Alps and on the La Réunion island. *Natural Hazards and Earth System Sciences Discussions*, 1-27. <https://nhess.copernicus.org/preprints/nhess-2020-322/>

•Bourrier, F., Toe, D., Garcia, B., Baroth, J. & Lambert, S. Experimental investigations on complex block propagation for the assessment of propagation models quality. *Landslides* 1612–5118 (2020).

Please, reformulate this sentence.

3) Line 34: Important references that details the steps and recommendation for the quantitative analysis of landslide and rockfall risk:

•Corominas J, van Westen C, Frattini P, Cascini L, Malet JP, Fotopoulou S, Catani F, Van Den Eeckhaut M, Mavrouli O, Agliardi F, Pitilakis K, Winter MG, Pastor M, Ferlisi S, Tofani V, Hervás J, Smith JT (2014) Recommendations for the quantitative analysis of landslide risk. *Bull Eng Geol Environ* 73:209–263

•Ferrari, F., Giacomini, A., Thoeni, K., & Lambert, C. (2017). Qualitative evolving rockfall hazard assessment for highwalls. *International Journal of Rock Mechanics and Mining Sciences*, 98, 88-101.

4) Line 63-64: "Although these initial studies demonstrate the possibilities of new experimental techniques, they still do not have the statistical basis, i.e. number of experiments, range of rock types, etc., to quantify energy dissipation rates and therefore calibrate modelling tools."

Yes, I doubt that it is possible to design an experiment that will be statistically significant in rockfall propagation (even for a specific site): it would cost so much and take so much time! This very study with "only" 82 bloc releases should not be considered very significant considering the high variability of travel

paths (even if it is an honorable score). However, each of these studies contribute to tackle these key aspects of rockfall propagation. Maybe you should rephrase by stating that this study is an additional contribution on the subject...with some specificities.

5) Lines 160-163: feedback would be welcome on the various instrumentation failures (which concerns about 40% of the tests). In addition, the post-processing procedure is not enough detailed as well as the artefacts encountered in the signals (also in Figure 3).

6) Equation (1): specify the variables w and m .

7) Figure 4 and Figure 5: the validity range of your empirical laws should be more discussed. I am skeptical about your extrapolation toward small masses, as you do not have control points (and whether such "shapes" are physically meaningful...). The quality of the fitting laws ought to be described with some metrics and the values of the solved constants ought to be described with their standard deviation. For example, how strongly is established the difference of behavior between cubic and platy boulders (lines 205-207, lines 229-231)?

8) Line 224-226: Since some of your results might not apply in other site configuration (channeling, topographical steps, different lithology, etc.), authors should state perspectives on their work.

Response to REVIEWER COMMENTS

Reviewer #1 (Remarks to the Author):

This paper presents results from an extensive campaign of block propagation experiments. The experimental procedure and, in particular, the measurement of block trajectories and kinematics changes during interaction with the ground were done using advanced techniques. This paper is of interest for the scientific community in the field of rockfall hazard given the amount of data acquired and the quality of the measurements done. In addition, this data is freely available in an Envidat repository. For these reasons, I recommend publication of the paper.

I don't have specific comments as regards to the introduction and study site description since both sections are very well written and show the expertise of the authors as regards to this topic. However, I also think that several points have to be improved in particular concerning the description of the details of the measurements, results analyses and the discussion. For that reason, I recommend major revision of the paper.

Thank you for taking the time to read our manuscript and for judging our work worthy of publication. The raised concerns and suggestions for revisions have been carefully addressed and incorporated in the revised manuscript.

The following major points have to be clarified / improved :

1) Experimental procedure : the initial conditions are not detailed although they are major causes of the propagation variability. These conditions should be detailed in terms of initial release height, block initial orientation, procedure used to initiate propagation. The variability of these conditions could also be quantified.

A: The release point is defined by the hydraulic platform installed at the top of the slope. Each rock is placed on the exact same location (within the rocks dimensions) and released via hydraulic tilt of the platform. This start conditions from zero velocity to a smooth slide off the platform at the exact same platform are quasi-identical apart from the geometrical orientations. The initial conditions are specified in greater detail in the campaign description to accommodate these facts.

2) Measurements : end point and scarring point locations. The authors present a 5 cm resolution that is very low according to me given that these points have to be estimated by the operator. Could the authors provide information on this point

A: The mentioned accuracy only applies for in-field measurement of endpoint locations and mapped scars for reference purposes and is the equipment accuracy under optimal handling. In order to clarify the in-field measurements – as opposed to a-priori impact mapping – the clarification “measure endpoint and scarring locations on the slope “ is added.

3) Measurements: the authors used topography, cameras, accelerometers and gyroscopic sensors. However, the role of the different measures in the analysis is not clearly described (for example, were the trajectories reconstructed in 3d using only the cameras or also using the locations of scarring points ?). Could this point be clarified ?

A: Reconstruction is performed in an ideal case via matching temporal information about impact and lift-off taken from the sensor streams combined with scarring locations taken from differential UAS surface models. We added this information to clarify the procedure to the reader. We will add an detailed overview table in the Envidat data set in order to make clear for potential users how the reconstructions were performed.

4) Figure 3 : I think that the time evolutions of angular velocities (upper rights) are examples for one block of each size. It may be specified. How variable are these time evolutions from one block to another ?

A: Correct. The inset displays angular velocities for the four measured weight classes and two shape classes. The variability can be inferred from the former Figure 4, the individual plotted gyroscopic data streams, but rather from the boxplot summary. With the resubmission, we revised Figure 4 completely in order to present more insights into different block sizes and shapes. The new Figure 4 highlights the differences in three-dimensional gyroscopic behavior between shape and weight classes and keeps the summary boxplot as comparison measure. Additional information about energy ratios and wobbling frequencies are also supplied in the new Figure 4.

5) Equation 1 : it seems to me that this relationship was established using all data per rock type and rock mass. So, it is a « mean relationship ». Could the authors clarify the meaning of this « mean relationship » given that, in Fig. 3, it is shown that the angular velocity evolves along the trajectory of each block ?

A: Preceding work (as the Induced Rockfall Dataset (Small Rock Experimental Campaign), Tschamut, Grisons, Switzerland data set (10.16904/engvidat.37), or other gathered experimental data) led to the conclusion that those slope specific bell curve evolutions of angular velocity do serve well as calibration variable. In order to collapse it to a single value, the mean relationship is found to be a fast, accessible benchmark value. The manuscript was amended where we introduce this mean relationship to make its purpose clearer.

6) Equation 2 : Could the authors specify how the kinetic energy changes were calculated. Only the translational velocities were used ? How did the authors determine the velocity « before interaction » and the velocity « after interaction » ? What is the definition of an « interaction » ?

A: The parabolic sections resulting from impact reconstruction always range from specified lift-off to subsequent impact location at the differential digital surface model. All parameters of interest (POI) such as position, kinematic variables, StoneNode readings such as angular velocities and acting accelerations are extracted at any given reconstruction point and allow to derive secondary variables such as translational and rotational energy. Impact analysis consists thus of extracting POI at the end point of one parabolic section and the lift-off POI of the subsequent parabolic section. Each impact – or interaction - correspond spatially to the line from impact to lift-off location, the ground scar. The manuscript was amended with the above explanation of the procedure and interaction definitions.

7) Figure 4 and Figure 5 : all the plots of the statistical relationships (small graphs) are plotted for mass values ranging from almost 0 to more than 50000 kg although the larger mass of the rocks used is 2650 kg. I think that the range of the plot should be reduced to the range of the blocks released.

A: The revised figures contain shortened axis ranges adapted to the order of magnitudes also used in the field experiments.

8) Figure 5 c : the positive values of the kinetic energy changes during impact suggest increases in kinetic energy. It can be explained if only translational velocities are used. This interesting point has to be discussed.

A: The occasional impact with a positive net energy intake can be attributed to a almost fully elastic rebound with a considerable potential energy intake from the altitude change between impact and lift-off position. We added a clarification of this process along with an uncertainty estimation on the reconstructed velocities.

9) Figure 5 c : the changes in rotational velocities during interaction are almost not described. It could be interesting to try to interpret these changes, especially in relation to changes in kinetic energy during impact.

A: In order to make changes of rotational energies better visible, Figure 5c was revised to plot the absolute values of each rotational energy change. The figure highlights the scaling of rotational changes with mass. Additionally, with the revision of Figure 4, a Energy ratio plot was introduced to highlight the rotational vs. translational energy components and its behavior during the descent. It also gives an estimate on rotational energies of projectiles impacting flexible barriers.

9) p.6 l 194-200 : I don't understand.

A: The data corroborates the intuitive and empirical fact, that smaller and slower rocks do produce smaller scars upon impact. The scarring effect is governed by the soil composition (compressibility, thickness as leading factors, moisture content, vegetation content, etc. as subsidiary factors) and the impact configuration of the projectile (mass, incoming velocity). The revised manuscript has been amended to state this idea of soil compaction in a more concise way.

10) Conclusion : it must be specified that the relationships obtained are site-specific and should be confirmed on other sites ?

A: Generally, the site specificity of the results has been incorporated throughout the manuscript, but is equally rephrased within the conclusion.

11) Conclusion : I agree that the paper emphasizes the « effect of block shape on propagation » but the effects shown in this paper could be recalled and summarized here ?

A: We agree with the reviewer, that a short summary of the findings rounds off the manuscript. An additional paragraph has been added before the final conclusions.

Minor points :

1) p.6 l.196 : “In an analogous”

A. resolved.

2) Figure 4 b : typo

Reviewer #2 (Remarks to the Author):

The paper presents the results of an extensive rockfall testing campaign conducted at the Chant Sura site in Switzerland. I find the paper well written and containing a unique large data set of full scale rockfall testing results.

However, a more comprehensive analysis of the results and some further discussion about their application for hazard assessment purposes could have been provided.

We thank the Reviewer #2 for the constructive comments and feedback. In the following the original comments of the Reviewer #2 are listed in bold italics and replied point-by-point.

The introduction summarises some of the numerous studies conducted over the last four decades in the scientific literature. However, some pioneering research on rockfall impact/rebound and energy loss at impact should also be considered.

The experimental set up and methodology for the reconstruction of trajectories have already been extensively presented in [37]. While this is one of the most interesting aspects of the proposed work, the full trajectory reconstruction of falling blocks by video fotogrammetry (including the full set of dynamic parameters at impact and rebound) is not new, and it has already been achieved by several researchers in recent years (see Dewez et al., 2010; Mathon et al. 2010; Giacomini et al. 2012; Volkwein et al. 2018 to name a few).

A: We agree with the reviewer, that the deployed experimental means should be seen as continuous adaptation and extension of trajectory reconstruction work in the past. We also do not consider the experimental methodology itself as main contribution, it merely serves as tool in order to obtain the presented data set. Thus, we consider the adoption and repetitive execution to a full slope scale, covering larger distances on relatively unobstructed terrain - resulting in the presented data set – as the key element of the presented work – in line with Reviewer 1 and 3. We gladly incorporate pioneering work of Dewez et al., 2010. We prefer citing the full report than merely conference proceedings, as the quality checks for most conference proceedings are - in our humble opinion - to low in order to account as a proper reference – and as much as we love the RSS conference series – we do not exclude their procedures from this assessment.

Accessing the report – available only in French – we deeply commiserate that such pioneering work is basically hidden of the community and requires non-standard searching approaches in order to reach the BRGM depository.

While we agree, that the trajectory reconstruction is not new, the extent of trajectory reconstruction provided by different studies in the past is often limited on a small numbers of rocks and/or rebounds, that is limited falling distances. Nonetheless, the extension to full slope scale as presented by Bourrier et al. 2020 and within this work is a major progress as experimental logistics and data treatment increase significantly.

We accommodated additional predecessor work of significant quality to the reference list, Dewez et al. 2010, more prominently to make the limited contribution in our hands to highlight this shadow banned, pioneering work.

Similarly, the generation of high-resolution DSM by aerial photogrammetry, and the post processing of data using a commercial software such as PhotoScan are becoming very common practice in remote sensing applications for rockfall analyses (see recent works published on the topic in various international journals in the field of remote sensing).

A: We absolutely agree with the reviewer. We only intended to give the reader a full, concise methodology setup not requiring the reader to search for additional experimental specifications elsewhere. We see the UAS methodology as mere tool to produce our own high-quality input data and thus also desisted to elaborate in a separate methods sections as in line with Reviewer 2, we do not see the tools deployed as the major contribution of this work as rather is the produced data set and its applicability for the community.

Recent work on the consideration of block shape has also been conducted on the numerical investigation of the shape effect on rockfall motion (see Ji et al.2019, Yan et al. 2019).

A: Work of Yan et al. have already been recognized in the initial submission, as their proposed model is conceptually similar to our own model (Leine et al. 2014,2020) which we are continuously refining since several years. Our intention with this work is to present valuable and complete data to the hazard assessment community irrespective of the pursued model approach. Nonetheless, we highlighted on-going work for inclusion of arbitrary shape in modern simulation tools more prominently in the manuscript.

The tendency of wheel shaped rocks to stabilise around their largest moment of inertia has already been highlighted by previous researchers ...

A: We do agree with the reviewer and also did not claim those findings as new. The stabilization around the largest moment of inertia or the so called intermediate axis theorem is a well-known fact. However, its physical manifestation as well as the experimental validation especially in the context of rockfall has been scarce and with the StoneNode measurements in particular, we present a vast data basis for this effect.

and observations about “stiletto effect”, or simply stress concentration, related to the “bullet effect” is also not new (see [38] and related works).

A: Our intention is to present data backed equivalence for the bullet effect (as termed by Spadari et al., 2018, for stress concentration on flexible rockfall barriers) for ground stress concentration imposed by rock-soil interactions. As the stress concentration on soil rather punches the ground than penetrates through it, we found stiletto effect a worthy representation and homage to the bullet effect.

I appreciate the significant number of tests conducted in the research and I acknowledge that it is, indeed, a very valuable and unique data set. However, the observations collected in terms of flight paths, associated velocities, and measured spreading angle are site specific. The empirical power law proposed for the angular velocity is, therefore, site dependent only for future numerical calibrations.

A: Generally, the site specificity of the results has been incorporated throughout the manuscript, but is equally rephrased within the conclusion.

The authors conclude that the spreading angle definition after release could be exploited in rockfall hazard assessment, however, no methodology is proposed for this purpose.

A: As we argue for incorporation of shape considerations in hazard scenarios – where applicable – we rephrased this statement more clearly in the discussion.

Graphs included in Figure 3 are hardly readable. Similarly, Figure 4, results very difficult to read because of the small fonts used in the graphs. Additionally, its significance (StoneNode outputs are reported for the entire experimental campaign) is also quite unclear.

A: Figure 3 is amended for better readability. Figure 4 has been substituted with a complete new figure. The significance of the presentation of the entirety of available StoneNode data set was to enable the reader to verify impact acceleration and rotational behavior through raw measured data, rather through a processed data plot. However, we do agree that this is obsolete to a certain point, as this verification can be done by any interested reader via the available data set. The figure will be available in the data set but omitted from the manuscript.

The presentation of the results is quite narrow and, given the extensive data set, a more thorough analysis could have been conducted.

A: While we don't feel like the presented analysis does not consist of thorough work, we take the reviewer's opinion very serious and added several new analysis aspects to the revised manuscript. With the completely revised Figure 4 we aim for a tangible representation on rotational behavior throughout the weight and shape classes. The different kinematic regimes become visible by comparison of the different gyroscopic streams. The summary panel and generalized – site specific- mean rotational relationship were carried over but presented more clearly. Additionally, their value as collapsed benchmark number is discussed in the manuscript. While manifestations of larger moment of inertias are common sense, its experimental manifestation – especially when being linked to concrete values – is in our opinion of large value to the community. On this track the wobbling frequency component where extracted as well.

The availability of rotational information allows to quantify the energy ratio between translational and rotational energy. This is a measure of importance, where real world data has been merely inexistent up to now. It can serve for example as indication on what additional percentage of energy dissipation must be taken into account for rockfall barriers.

The presented data set consists of a plethora of information and the presentation was also intended to serve as an overview for the data set. We do agree that the data set can be exploited further

Reviewer #3 (Remarks to the Author):

The work of Caviezel et al. studies the results of a block release experiment that was documented in detail (sensors inside the blocks, high-resolution video records, etc.). Although it is not the first publication on this subject by this scientific team, their experimental set up is rather original and allow gathering new information on the mechanisms at stakes during rockfalls. I find this work relevant for publication.

I welcome the effort of transparency and contribution to the scientific community by making available the data collected during this experiment.

-Introduction:

-Stat of the art: Some references are missing (see minor comments)

A: Missing references are added.

-The authors should be more specific about their contribution in rockfall hazard assessment (see minor comments)

-Experimental set up: OK

-Methodology used to analyze the results of the experiment: on the overall, it is OK. Some explanations are missing (see minor comments).

-Discussion: on the overall, it is OK (see minor comments).

Minor comments:

1) overall remark:

I appreciate the enthusiasm of the authors who want to improve practices in rockfall hazard assessment. However, I would like them to express more specifically their contribution to this scientific field, especially in the abstract and the introduction. Their work provides concrete elements to improve the calibration of propagation models (as they state in the discussion), which represents only one step among many in rockfall hazard assessment. Even in this specific field of assessing rockfall propagation, research is still very active on documenting other physical phenomena (fragmentation of blocks, interactions of blocks during propagation, etc.) which are not mentioned here.

2) Line 20: "Presently, there exists no experimental basis on how rockfall hazard scales with rock mass, size and shape": There is very scarce work on the subject. However, some works were recently published on the subject:

•Wegner, K., Haas, F., Heckmann, T., Mangeney, A., Durand, V., Villeneuve, N., & Becht, M. (2020). Assessing the effect of lithological setting, block characteristic and slope topography on the runout length of rockfalls in the Alps and on the La Réunion island. *Natural Hazards and Earth System Sciences Discussions*, 1-27. <https://nhess.copernicus.org/preprints/nhess-2020-322/>

•Bourrier, F., Toe, D., Garcia, B., Baroth, J. & Lambert, S. Experimental investigations on complex block propagation for the assessment of propagation models quality. *Landslides* 1612–5118 (2020).

Please, reformulate this sentence.

A: We reformulated the abstract part and added the newest references. Bourrier et al. was already referenced, but we added a introductory part about shape considerations.

3) Line 34: Important references that details the steps and recommendation for the quantitative analysis of landslide and rockfall risk:

•Corominas J, van Westen C, Frattini P, Cascini L, Malet JP, Fotopoulou S, Catani F, Van Den Eeckhaut M, Mavrouli O, Agliardi F, Pitilakis K, Winter MG, Pastor M, Ferlisi S, Tofani V, Hervás J, Smith JT (2014) Recommendations for the quantitative analysis of landslide risk. *Bull Eng Geol Environ* 73:209–263

•Ferrari, F., Giacomini, A., Thoeni, K., & Lambert, C. (2017). Qualitative evolving rockfall hazard assessment for highwalls. *International Journal of Rock Mechanics and Mining Sciences*, 98, 88-101.

A: We included the references.

4) Line 63-64: "Although these initial studies demonstrate the possibilities of new experimental techniques, they still do not have the statistical basis, i.e. number of experiments, range of rock types, etc., to quantify energy dissipation rates and therefore calibrate modelling tools."

Yes, I doubt that it is possible to design an experiment that will be statistically significant in rockfall propagation (even for a specific site): it would cost so much and take so much time! This very study with “only” 82 bloc releases should not be considered very significant considering the high variability of travel paths (even if it is an honorable score). However, each of these studies contribute to tackle these key aspects of rockfall propagation. Maybe you should rephrase by stating that this study is an additional contribution on the subject...with some specificities.

A: We agree with the reviewer, that statistical significance with respect to rockfall experiments is very hard to achieve. We believe, however, that with artificial, reproducible rocks, we have achieved an unprecedented systematics and that these 183 released rocks and 82 reconstructed trajectories with 1394 impacts indeed can serve as calibration basis, especially when focusing on impacts on different soil classes. We added an according grading of the presented study in the discussion.

5) Lines 160-163: feedback would be welcome on the various instrumentation failures (which concerns about 40% of the tests). In addition, the post-processing procedure is not enough detailed as well as the artefacts encountered in the signals (also in Figure 3).

A: Data availability is not equal to the failure rate as the availability was predominantly governed by the available number of sensors, as they were developed during the first part of the experimental series.

6) Equation (1): specify the variables w and m .

A: The parameters were specified

7) Figure 4 and Figure 5: the validity range of your empirical laws should be more discussed. I am skeptical about your extrapolation toward small masses, as you do not have control points (and whether such “shapes” are physically meaningful...). The quality of the fitting laws ought to be described with some metrics and the values of the solved constants ought to be described with their standard deviation. For example, how strongly is established the difference of behavior between cubic and platy boulders (lines 205-207, lines 229-231)?

A: The validity range is reduced to the empirical data. A remark about extrapolation and site specificity is added in the text. As the plots were already showing the standard deviations of the fitted curves in the initial submission, their standard deviations are added additionally to the text.

8) Line 224-226: Since some of your results might not apply in other site configuration (channeling, topographical steps, different lithology, etc.), authors should state perspectives on their work.

A: Generally, the manuscript has been revised to stress site specificity more. Additionally, perspectives on the results, their deviations and adaptations as well as interesting site configurations are added in the discussion.

Reviewers' Comments:

Reviewer #1:

Remarks to the Author:

The authors substantially improved the paper to answer the points raised by the reviewers. I do not have additional requirements and recommend to accept the publication of the paper.

Reviewer #2:

Remarks to the Author:

The paper has been significantly revised. Substantial modifications have been included and data better analysed according to the reviewer's suggestions.

However, I think there is still some work to be conducted to bring the manuscript at the standards of the journal. There are several small grammatical errors all along the manuscript and I recommend using English professional services to review the entire manuscript.

Some examples:

- Line 176: "number of sensors" instead of "amount of sensors"?
- Line 214: "each impact corresponds" instead of "each impact correspond"
- Line 248: "major uncertaintyarises" instead of "major uncertaintyarise"
- Line 277: shape dependence "is" instead of "are"
- Paragraph between lines 174-180- check English and meaning of the sentences. It is not clear.

Some additional comments to address:

- Fig 4a is too small and difficult to read, I suggest increasing the size of each sub-figure
- Lines 223-225: the authors suggest "the extrapolation and validity of the proposed empirical fitting laws must be verified in further studies". I recommend more specific details about what would be required to validate the approach. Do the authors suggest considering another experimental campaign conducted under similar controlled conditions? Would it be possible to use some results in the scientific literature that follow a similar data collection approach (data can be found in the literature, even if with much smaller datasets) to validate the proposed empirical fitting?
- Lines 233-235: the first part of the sentence does not really justify the meaning of the observation included in its second part. This point should be clarified. Larger and faster blocks can produce a variety of scars (not always bigger) depending on the soil conditions and soil type.
- Lines 245-248: the authors attribute the occasional impact with positive "empirical normalised energy dissipation" to special elastic rebound conditions. Given the extensive data set available, this could be investigated by looking into the "impact angle" and "rebound angle". I suggest analysing this data to reinforce the observations.
- My previous comment also relates to the results shown in Figure 5c and 5d. Figure 5d is particularly interesting and I suggest looking into impact velocity, impact angle and rotational velocity at impact to provide further interesting insights. The authors agree that the data can be exploited further, so I think there is a valid ground for additional analyses to reinforce the manuscript.
- Finally, the authors refer to "code availability". Can you please provide more details? (e.g. code name, computing requirements)

Response to REVIEWER COMMENTS

Reviewer #1 (Remarks to the Author):

The authors substantially improved the paper to answer the points raised by the reviewers. I do not have additional requirements and recommend to accept the publication of the paper..

Once again, we thank Reviewer #1 for taking the time to read our revised manuscript and for judging our work worthy of publication.

Reviewer #2 (Remarks to the Author):

The paper has been significantly revised. Substantial modifications have been included and data better analysed according to the reviewer's suggestions.

However, I think there is still some work to be conducted to bring the manuscript at the standards of the journal.

We thank the Reviewer #2 for the additional comments. In the following the comments of the Reviewer #2 are listed in bold italics and replied point-by-point.

There are several small grammatical errors all along the manuscript and I recommend using English professional services to review the entire manuscript.

A: The manuscript has been proof read by native English speaker from the beginning. An additional round of proper proof reading has been conducted.

Some examples:

- *Line 176: "number of sensors" instead of "amount of sensors"?*
- *Line 214: "each impact corresponds" instead of "each impact correspond"*
- *Line 248: "major uncertaintyarises" instead of "major uncertaintyarise"*
- *Line 277: shape dependence "is" instead of "are"*

The revised manuscript has been changed to correct third person singular verb forms alongside with the correction of some other typos.

- *Paragraph between lines 174-180- check English and meaning of the sentences. It is not clear.*

A: The paragraph has been clarified, to better explain the reasoning for sensor failure.

Some additional comments to address:

- *Fig 4a is too small and difficult to read, I suggest increasing the size of each sub-figure.*

A: We can relate to the reviewer's concern. However, the presented figure, panel 4a in particular has been meticulously curated to display the sensor streams in their correct ratio and time length alongside with the analysis panels 4b-e. In the age of vector graphics, we believe that presenting the picture as is, in high quality, being able to use and display digitally in zoomed version remains adequate and in line with the journal's history of high information density graphics.

- *Lines 223-225: the authors suggest "the extrapolation and validity of the proposed empirical fitting laws must be verified in further studies". I recommend more specific details about what would be required to validate the approach. Do the authors suggest considering another experimental campaign conducted under similar controlled conditions? Would it be possible to use some results in the scientific literature that follow a similar data collection approach (data can be found in the literature, even if with much smaller datasets) to validate the proposed empirical fitting?*

A: Both proposed methods are a viable approach. Clarification has been introduced in the manuscript. The use of scientific literature/data is dependent on the availability of the temporal information in the trajectory reconstruction and also in the availability of the raw data.

- *Lines 233-235: the first part of the sentence does not really justify the meaning of the observation included in its second part. This point should be clarified. Larger and faster blocks can produce a variety of scars (not always bigger) depending on the soil conditions and soil type.*

A: The sentence has been rectified to link the penetration depth to the incoming velocity as described by the referenced literature. The scar dependence upon impact configuration etc. has been described in the present manuscript already in the subsequent sentences.

• Lines 245-248: the authors attribute the occasional impact with positive “empirical normalised energy dissipation” to special elastic rebound conditions. Given the extensive data set available, this could be investigated by looking into the “impact angle” and “rebound angle”. I suggest analysing this data to reinforce the observations.

A: We did perform additional analysis of impact and rebound angle in order to scrutinize these “outliers” furthermore. However, no clear correlation is found which is also partly attributed to the limitations of the reconstruction workflow, also mentioned in the manuscript. Although the level of detail in the data is very high, the resolution of the exact impact configuration with respect to impact geometry (rock orientation and ground meso-scale surface roughness) is not available to that extent to draw reliable conclusions. We added clarification of this fact in L247 and proposed future solution with respect to sensor deployment in L. 292 ff.

• My previous comment also relates to the results shown in Figure 5c and 5d. Figure 5d is particularly interesting and I suggest looking into impact velocity, impact angle and rotational velocity at impact to provide further interesting insights. The authors agree that the data can be exploited further, so I think there is a valid ground for additional analyses to reinforce the manuscript.

A: As mentioned in the answer to the previous remark, that analysis has been performed but did not yield stringent results, mainly due to the lack of knowledge about the local impact geometry. The manuscript has been complemented with a statement with respect to the limitations of this data set as well as with a statement pointing towards future, more sophisticated machine learning analysis and the trial of suitably trained neuronal networks for the analysis of IMU data. And certainly enough, there is always room for more analysis, especially on large data sets like the one presented, but goes in our opinion beyond the scope of this manuscript.

• Finally, the authors refer to “code availability”. Can you please provide more details? (e.g. code name, computing requirements)

A: As the analysis was done with regular MATLAB routines and the provided data set is universally readable, we omitted the “Code availability” statement, as it is not applicable.